# *BRAF* Mutations in Colorectal Liver Metastases: Prognostic Implications and Potential Therapeutic Strategies

**DOI:** 10.3390/cancers14174067

**Published:** 2022-08-23

**Authors:** Pei-Pei Wang, Chen Lin, Jane Wang, Georgios Antonios Margonis, Bin Wu

**Affiliations:** 1Department of General Surgery, The First Affiliated Hospital of USTC, Division of Life Sciences and Medicine, University of Science and Technology of China, Hefei 230001, China; 2Department of General Surgery, Peking Union Medical College Hospital, Chinese Academy of Medical Sciences, Peking Union Medical College, Beijing 100730, China; 3Department of Surgery, University of California San Francisco, San Francisco, CA 94158, USA; 4Department of Surgery, Memorial Sloan Kettering Cancer Center, New York, NY 10065, USA

**Keywords:** colorectal cancer, liver metastases, *BRAF* mutation, prognosis, targeted therapy

## Abstract

**Simple Summary:**

In this literature review, we investigated the relationship between *BRAF* mutation and prognosis in patients with colorectal cancer liver metastases. We also investigated factors affecting the prognosis of patients with BRAF mutations and summarized the latest research on targeted therapies.

**Abstract:**

Surgery combined with chemotherapy and precision medicine is the only potential treatment for patients with colorectal cancer liver metastases (CRLM). The use of modern molecular biotechnology to identify suitable biomarkers is of great significance for predicting prognosis and formulating individualized treatment plans for these patients. *BRAF* mutations, particularly V600E, are widely believed to be associated with poor prognosis in patients with metastatic CRC (mCRC). However, it is unclear which specific factors affect the prognosis of CRLM patients with *BRAF* mutations. It is also unknown whether patients with resectable CRLM and *BRAF* mutations should undergo surgical treatment since there is an increased recurrence rate after surgery in these patients. In this review, we combined the molecular mechanism and clinical characteristics of BRAF mutations to explore the prognostic significance and potential targeted therapy strategies for patients with *BRAF*-mutated CRLM.

## 1. Introduction

Colorectal cancer (CRC) is the second most common cause of cancer-related death in the United States [1]. The liver is one of the most common sites of metastasis. Approximately 15–25% of patients diagnosed with CRC have liver metastasis at the time of diagnosis, and up to 50% develop metastasis within 3 years [2]. The overall 5-year survival rate of patients with surgically treated CRLM is 25–58%, whereas the 5-year survival rate of patients with unresectable liver metastases is only 10–15% [3]. Only 20–30% of patients with CRLM have resectable disease at initial presentation [4], and patients who undergo resection have a 50–60% chance of recurrence after surgery [5]. Surgery combined with chemotherapy, targeted therapy, and immunotherapy is the only potential treatment for CRLM. The use of modern molecular biotechnology to investigate the pathogenesis of CRC and find suitable biomarkers is of great significance for predicting prognosis and formulating individualized treatment plans. Specifically, molecular diagnosis is a key factor in the management of metastatic colorectal cancer (mCRC). *KRAS*, *NRAS*, and *HRAS* are the best-studied proteins of the rat sarcoma virus (*RAS*) subfamily [6] and are important molecular markers in CRC. For example, *RAS* mutations are associated with poorer overall survival (OS), disease-free survival (DFS), and relapse-free survival (RFS) [7]. Not surprisingly, molecular drugs targeting *KRAS* and *NRAS*, epidermal growth factor receptor (*EGFR*), and vascular endothelial growth factor (*VEGF*) have long been used as first-line treatments for mCRC [8]. Other genes, such as *TP53* and *APC/PIK3CA* are important genetic markers for evaluating tumor biology [9]. Drugs targeting these proteins are currently under development.

V-RAF mouse sarcoma B virus oncogene *(BRAF*), another member of the *RAS* family, is a protein kinase that plays a key role in the mitogen-activated protein kinase (MAPK) APK/ERK signaling pathway [10]. Abnormal activation of this pathway plays an important role in the development of CRC. Specifically, mutations in the *BRAF* gene result in the continuous downstream activation of MEK/ERK/MAPK pathways [11,12], which affects tumor cell differentiation, migration, and proliferation. The *BRAF* mutation rate is 4.7–20% in CRC and 1–6.1% in patients with resectable CRLM [13,14,15,16]. This may be due to the rapid tumor progression caused by *BRAF* gene mutation which renders these CRLM cases technically unresectable. The current consensus is that *BRAF* mutations are associated with poor prognosis in mCRC patients, with mortality nearly thrice that of patients with wild-type *BRAF*.

## 2. Molecular Mechanism

*BRAF* is located on human chromosome 7q, encodes serine/threonine protein kinases, and belongs to the *RAF* gene family along with *ARAF* and *CRAF*. *RAF* family members are composed of three conserved regions: CR1, CR2, and CR3 [17,18,19]. CR1 is a RAS GTP-binding self-regulating domain, CR2 is a serine-rich hinge region, and CR3 is a catalytic serine/threonine protein kinase domain. Among these, *BRAF* has the strongest kinase activity and is the most effective activator of MEK/ERK protein; it triggers the MAPK signaling pathway by activating downstream MEKs (MEK1 and MEK2) and ERKs (ERK1 and ERK2) (Figure 1). In comparison, *ARAF* and *CRAF* mostly act as housekeeping genes [20]. *BRAF* mutations can be subdivided into three classes. Class 1 is independent of RAS and includes independent active monomers, whereas Class 2 functions as active dimers. Both Class 1 and Class 2 *BRAF* proteins become largely independent from their upstream regulator, RAS GTPase, for growth and proliferation in cancer [21]. In contrast, Class 3 *BRAF*-mutated proteins rely on RAS signaling for maximal activation, which means their activity depends on RAS-GTP levels. Therefore, blocking upstream RAS signaling is a potential treatment strategy for CRC with Class 3 *BRAF* mutations, and also provides a theoretical basis for the stratified treatment of *BRAF* mutations [22,23]. Specifically, V600E-mutated CRC is highly malignant and lacks effective treatment, resulting in a poor prognosis. V600E mutated CRC has recently been divided into two distinct subgroups: BM1 and BM2. The former exhibits high *KRAS*/mammalian rapamycin (mTOR)/AKT/4EBP1 expression, EMK activation, and immune infiltration, whereas the latter exhibits disordered cell cycle checkpoints [24,25]. Importantly, drug screening tests suggest that different subtypes of V600E may respond differently to specific drugs, such as *BRAF* and MEK inhibitors. The classification and types of *BRAF* mutations are shown in Figure 2.

V600E, which is a Class 1 mutation where glutamic acid replaces valine [26], accounts for approximately 90% of *BRAF* mutations in CRC [9]. As mentioned above, Class 1 mutations enable constitutive activation of MAPK signaling pathways independent of RAS. V600E mutations are usually mutually exclusive of *KRAS* and *NRAS*, suggesting that changes in MAPK signaling alone are sufficient to induce tumorigenic activity and are an important oncogene in CRC. The proportion of non-V600E mutations, which are Class 2 and 3 mutations, is less than 10% and their clinical incidence is low [27]. The biological characteristics of CRC caused by such mutations are different from those with V600E mutations and should be distinguished in basic and clinical studies.

## 3. Clinicopathological Features

*BRAF*-mutated CRC is more likely to occur in the elderly and women, mostly occurs in the right colon, and is prone to liver, peritoneum, and distant lymph node metastasis [28,29,30,31]. The pathological features of the tumor include poor differentiation, increased mucinous adenocarcinoma components, and high aggressiveness. Specifically, *BRAF* mutations are drivers of serrated polyp pathways, which are considered precursors to CRC [32]. Furthermore, *BRAF* mutations may be related to high microsatellite instability (MSI-H)/mismatch repair deficient (dMMR) status, as they are more common in patients with dMMR. Of note, patients with *BRAF* mutations/MSI have good prognoses, followed by *BRAF* wildtype/MSS cancers, whereas patients with *BRAF*-mutated/MSS have the worst prognoses [33,34].

## 4. Clinical Implication

Through a literature review, we investigated the following four aspects: (1) the relationship between *BRAF* mutation and prognosis in patients with CRLM; (2) factors affecting the prognosis of patients with *BRAF*-mutated CRLM; (3) whether patients with initially resectable, *BRAF*-mutated CRLM should undergo surgical treatment; and (4) whether the recurrence rate is increased in patients with *BRAF*-mutated CRLM after surgical treatment.

### 4.1. BRAF Mutation and Prognosis

*BRAF* mutations occur in only 1–6.1% of patients with resectable CRLM, whereas *KRAS* mutations occur in 30–40% of these patients [13,35]; thus, the number of CRLM patients with *BRAF* mutations reported in a single center is usually small. For example, a multicenter study from Italy with the largest number of cases enrolled prior to 2015 found only 12 *BRAF* mutations in 309 patients who underwent surgical resection of CRLM. In another example, a meta-analysis that same year found that only 74 studies reported overall survival associated with *BRAF* mutation status, and disease-free survival was explored in only one study.

Fortunately, studies investigating the correlation between *BRAF* mutations and prognosis for patients with CRLM have increased. A retrospective study conducted in 2016 included six patients with *BRAF*-mutated CRLM and found that *BRAF* mutations consistently predicted poorer time to relapse (TRR) and disease-specific survival (DSS) [35]. This suggests that *BRAF* mutations can predict poor outcomes and may help guide treatment decisions. Of note, as this was only a preliminary study, the type of *BRAF* mutation was not considered. Then, a 2018 study stratified mutations in CRLM patients with multicenter data from seven academic institutions participating in the International Genetic Consortium for Colorectal Liver Metastasis (IGCLM) [13]. They found that the V600E *BRAF* mutation was associated with poor prognosis and an increased risk of recurrence. Interestingly, it was also found to be the strongest prognostic determinant in the cohort. In another example, a multicenter study of 1497 CRLM patients found that the median RFS was 22 months for patients with wild-type *BRAF* and 10 months for patients with *BRAF* mutations (*p* < 0.001) [36]. Finally, a 2019 retrospective study of 24 medical centers matched 66 patients with *BRAF* mutations with 183 patients with *BRAF* wild-type who underwent CRLM resection [37]. The 1- and 3-year overall survival rates were 94% and 54%, respectively, in the *BRAF* mutant group and 95% and 82%, respectively, in the *BRAF* wild-type group (*p* = 0.004). The median survival after disease progression was 23.0 months in the *BRAF* mutant group and 44.3 months in the *BRAF* wild-type group (*p* = 0.050).

Poor survival was observed after hepatectomy in *BRAF* V600E-mutated CRLM patients from Japan in 2020 (RFS: 5.3 months, OS: 31.1 months) [38]. A multicenter study in France that genotyped the tumors of 246 patients with wild-type *RAS* CRLM also found a strong association between *BRAF* mutations and poor survival. Specifically, *BRAF* mutations increased the risk of death by three times in patients with CRLM, with a median OS of less than 1 year. Similarly, a recent retrospective study in China included 492 patients with mCRC (280 with synchronous and 212 with asynchronous metastasis) [39]. Multivariate analysis suggested that *BRAF* and *NRAS* mutations were independent prognostic factors affecting OS. Another study in August 2021 included 63 patients with wild-type *BRAF* and 6 patients with *BRAF* mutations in the Amsterdam Liver Met Registry (AmCORE) [30]. The *BRAF* mutation group had significantly poor OS (*p* < 0.001), although there was no statistically significant difference in DFS between *BRAF* wild-type and mutant types (*p* = 0.075). Furthermore, *BRAF* V600E mutation status was a major determinant of OS in patients with CRLM in a multicenter retrospective study from China (*p* < 0.05) [40]. Finally, a recent meta-analysis evaluated the effect of *BRAF* mutant status on OS and DFS in CRLM patients [41]. A total of 1857 patients with known *BRAF* status were enrolled in the study. The results suggested that the OS and DFS of *BRAF*-mutated patients are significantly worse than those of patients with wild-type tumors. We have summarized all recent studies reporting survival outcomes of patients with *BRAF* mutation in Table 1.

### 4.2. Prognostic Risk Factors in Patients with BRAF-Mutated Tumors

*BRAF* mutations were observed in 35 of 1497 patients with CRLM enrolled in a multicenter study, 71% of whom had V600E mutations. Study results suggested that the OS of patients with *BRAF*-mutated CRLM was worse in those with positive primary tumor lymph nodes, embryonic antigen (CEA) > 200 mg/L, and concurrent tumor metastasis. A recent retrospective study similarly concluded that mCRC patients with synchronous metastases had poor OS compared to those with metachronous metastases. Of note, the small sample size of most studies that examine *BRAF*-mutated CRLM does not allow for meaningful statistical analyses to investigate which factors affect prognosis. Such analyses could be used to define selection criteria for surgery, as the value of surgery in patients with technically resectable *BRAF*-mutated CRLM has been questioned.

### 4.3. Effect of Surgical Treatment on Prognosis

Many scholars suggest that surgical treatment is not recommended for patients with CRLM with *BRAF* V600E mutations given their poor prognosis. In fact, a 2020 multicenter study from Japan suggested that systemic chemotherapy followed by hepatectomy in responders should be considered for patients with V600E mutations, even in upfront resectable cases [38]. Another contemporary study from Japan even suggested that patients with *BRAF*-mutated CRLM should be considered oncologically unresectable regardless of technical resectability [42]. This suggestion stemmed from the fact that the median OS (17.2 months) of the 28 patients with surgically treated V600E mutated CRLM was similar to that of the 28 patients with unresectable *BRAF*-mutated CRLM. The median OS of only 17.2 months for patients with resected V600E-mutated CRLM is lower than that reported by other studies for similar cases. For example, a recent multicenter retrospective study from France of 49 CRLM patients with surgically treated *BRAF* V600E-mutations s [43] reported a median OS of 34 months (28.9 to 67.3 months). Similarly, Margonis and colleagues [44] have reported a median OS of 30.6 months in 182 patients with surgically treated BRAF V600E-mutated tumors. Given that the updated analysis of the TRIBE-2 trial reported a median OS of 13.4 months for patients with unresectable BRAF-mutated CRLM, it is hard to deny surgery to patients with otherwise resectable CRLM on the basis of a BRAF V600E mutation alone [45]. Nevertheless, to definitively address this topic, we need properly matched comparisons of patients treated with resection vs. systemic therapy alone, especially to account for the advanced tumor burden of medically treated cases. This may reveal whether *BRAF* mutation should be a biologic contraindication for otherwise resectable *BRAF*-mutated CRLM.

### 4.4. Recurrence Rates of Patients Who Underwent Surgical Intervention of CRLM

A multicenter retrospective study from France suggested that *BRAF* mutations did not increase the risk of recurrence after surgery [37]. In contrast, a multicenter study in Japan of patients with *BRAF* V600E mutated CRLM showed that they were prone to early recurrence and had a very low survival rate after tumor recurrence after hepatic metastasectomy [38]. A meta-analysis confirmed a higher rate of liver and extrahepatic recurrences in patients with *BRAF* mutation who underwent liver surgery for CRLM [31]. Interestingly, a multicenter retrospective study of 47 patients with *BRAF*-mutated CRLM showed that although *BRAF* mutations were associated with poor outcomes early after surgery, patients with *BRAF*-mutated CRLM who survived the first year after surgery had similar outcomes as wild-type *BRAF* patients. This likely reflects the fact that patients with *BRAF* mutations and truly adverse disease biology (i.e., *BRAF* V600E mutations +/− other unknown modifiers of biologic aggressiveness) largely die of disease within the first postoperative year, whereas the survivors represent a far more favorable risk sub-cohort (e.g., patients with *BRAF* non-V600E mutations).

## 5. *BRAF* Inhibitors and Potential Targeted Therapies

In Ref. [46], *BRAF* mutations have been reported in 8–12% of mCRC cases, and mCRC patients with V600E mutations have particularly poor prognoses. Interestingly, mutations in *BRAF* V600E are also common in malignant melanoma and papillary thyroid carcinoma [47,48]. Additionally, although BRAF mutation has a negative prognostic role in papillary thyroid carcinoma, the overall favorable prognosis of this malignancy limits the impact of BRAF on long-term patient outcomes [49]. Previous studies have focused on inhibiting this signaling pathway by blocking the activity of *BRAF* and MEK, and drugs targeting these pathways have achieved remarkable efficacy in the treatment of melanoma. In fact, the combination of the *BRAF* inhibitor encorafenib and the MEK inhibitor binimetinib has been approved as a first-line treatment in patients with *BRAF* V600E mutated melanoma in the US and Europe. Compared with malignant melanoma, CRCs have a strong adaptive feedback signaling network. Experiments have found that the inhibition of V600E leads to a decreased expression of the MAPK signaling pathway, resulting in a loss of expression of ERK, which inhibits the activation of the MAPK pathway. This loss of negative feedback leads to the activation of RAS and other RAF kinases (such as *CRAF*), resulting in *BRAF* inhibitor-resistant RAF dimers that bypass the action of *BRAF* inhibitors and restore MAPK pathway signaling. Interestingly, studies have shown that anti-*EGFR* therapy may sensitize previously *BRAF*-resistant cell lines to *BRAF* inhibitors. Specifically, *BRAF* and *EGFR* inhibition results in persistent inhibition of MAPK signaling and tumor growth. Therefore, the current treatment strategy for *BRAF* V600E-mutated mCRC is chemotherapy combined with targeted inhibitors.

A side effect of MAPK pathway inhibition is the overactivation of *EGFR*. It is not entirely clear why downstream MAPK is unable to intercept the enhanced *EGFR* signal after inhibition. This suggests that *KRAS* and *BRAF* mutations activate downstream ERK signaling in a partially autonomous manner. Of note, the notion that *KRAS* mutations operate independently of upstream signaling has been challenged by genetic mouse models of lung and pancreatic cancers, which showed that *EGFR* inhibition prevented the growth of *KRAS*-mutated tumors. Interestingly, cell lines in 2D cultures showed significant intercellular variability in ERK signaling responses. This is important because measurements at the population level ignore single-cell heterogeneity. Thus, monitoring the pharmacological response of single tumor cell ERK signaling to drugs will greatly improve our understanding of responses to targeted therapies. Patient-derived organoids (PDOs) are powerful in vitro 3D models that retain the histopathological characteristics of tumors in vivo, including patient-specific drug responses [50]. They provide a new approach to the development of antitumor drugs for CRC. Studies have suggested that in organoid models, enhanced upstream *EGFR* activity improves the signal transduction efficiency of the MAPK pathway in *KRAS* or *BRAF* mutations. Thus, *EGFR* inhibitors can block the MAPK pathway induced by *BRAF* mutations to suppress tumors.

The FOLFOXIRI regimen (oxaliplatin + irinotecan + folate + fluorouracil) combined with bevacizumab has become the recommended first-line standard regimen for the treatment of mCRC with *BRAF* mutations, and studies suggest that this regimen can improve remission rate and survival. Of note, the PICCOLO trial reported a detrimental effect of adding panitumumab to chemotherapy (i.e., irinotecan) in patients with BRAF-mutated mCRC [51]. However, other trials (e.g., CRYSTAL) have shown that the addition of anti-EGFR agents (i.e., cetuximab) to irinotecan was associated with a trend toward improved PFS (8.0 vs. 5.6 months, *p* = 0.87) and OS (14.1 vs. 10.3 months, *p* = 0.74) [52].

The regimen of dabrafenib + trametinib + cetuximab can be used as an alternative treatment and can improve overall and progression-free survival. Clinical trials of drugs targeting *BRAF* mutations in CRLM patients are ongoing, including *BRAF* inhibitors, MEK inhibitors, and monoclonal antibodies against *EGFR*. Most clinical trials include monotherapy, double therapy, and triple therapy. The toxicities of targeted therapies are shown in Table 2.

Early exploration of *BRAF*-targeted therapies has mostly focused on drug safety studies. In a phase Ib trial, patients with *BRAF* V600E mutated mCRC received a regimen of the selective RAF kinase inhibitor encorafenib combined with the *EGFR*-targeting monoclonal antibody cetuximab, with or without the PI3K-alpha inhibitor alpelisib [53]. A total of 28 patients received triple therapy and 26 received double therapy. The primary objective of this study was to determine the maximum tolerated dose (MTD) or recommended dose for phase II clinical trials, which was found to be 200 mg encorafenib and 300 mg alpelisib. None of the groups achieved MTD during the study period. Importantly, both cetuximab and encorafenib showed good clinical activity and tolerability in the treatment of *BRAF*-mutated mCRC, and the safety of both triple and double therapies was acceptable. Another phase I clinical trial compared the two-therapy *BRAF* inhibitor darafenib and *EGFR* monoclonal antibody panitumab (D + P), the three-therapy darafenib, MEK inhibitor trametinib, and panitumab (D + T + P), and the two-therapy trimeitinib and panizumab (T + P) [54]. The results suggested that patients treated with D + T + P had a 21% chance of complete or partial response. Response rates for D + P and T + P were 10% and 0%, respectively. There was no significant difference in the overall incidence of adverse events among the three regimens in terms of toxicity and safety. It is noteworthy that there were two deaths in the triple therapy group, one due to bleeding and the other due to unexplained causes, both of which were considered drug-related. Finally, there is a notable phase 3 trial: *BRAF* V600E mutant mCRC open-label, randomized, three-arm, phase III BEACON CRC trial [55]. They found that the overall response rate was 48% after triple treatment with *BRAF*, MEK, and *EGFR* inhibitor cetuximab. The median progression-free survival was 8.0 months, median OS was 15.3 months, and median follow-up time was 18.2 months. The safety and tolerability of triple therapy were acceptable. From the above-mentioned clinical studies, we conclude that the safety and drug toxicity of triple therapy are generally acceptable. However, the clinical use of triple therapy may lead to severe drug toxicity events, and drug use should be closely monitored. The future use of targeted nanocarriers (NCs) may reduce systemic toxicity by improving local delivery to the disease site [56]. A recent pre-clinical study showed that certain nanoparticles (i.e., cubosomes) successfully targeted overexpressed carcinoembryonic antigens (CEA) on colorectal cancer cells [57]. This is particularly important as other studies have suggested that CEA can be used in conjunction with ctDNA to allow for more precise recurrence risk stratification and guide-personalized adjuvant treatment of CRLM [58].

*EFGR* inhibitors are often limited in the treatment of CRC due to dermal toxicity and clinical manifestations of acne-like rashes, which are caused by the inhibition of the MAPK pathway. As mentioned earlier, *BRAF* inhibitors activate MAPK downstream of *EGFR*, which could theoretically combat the development of dermal toxicity. Recently, a phase 1 clinical trial was conducted to test the hypothesis that topical treatment with the *BRAF* inhibitor LUT014 improves *EGFR* inhibitor-induced skin toxicity [59]. Ten patients with mCRC developed acne-like rashes during cetuximab or panizumab treatment. Six patients reported an improved rash after local treatment with LUT014. Ultimately, LUT014 was shown to be safe and effective in improving rashes and provided indirect confirmation of the mechanism by which *BRAF* inhibitors can induce the abnormal activation of MAPK.

Patients with *BRAF* V600E-mutated mCRC reportedly have a low response rate to the *BRAF* inhibitor vemurafenib. Interestingly, the blockade of *BRAF* V600E by vemurafenib upregulates *EGFR* feedback, whereas cetuximab can block the activation of the *EGFR* signaling pathway. The SWOG S1406 study enrolled 106 patients with mCRC with the *BRAF* V600E mutation and randomly assigned irinotecan and cetuximab with or without vemurafenib [60]. Progression-free survival improved with the addition of vemurafenib (HR 0.50, *p* < 001), and the disease control rate was 65% versus 21% (*p* < 0.001). Ultimately, we believe that triple therapy (*EGFR* and *BRAF* inhibitors combined with irinotecan) is effective in mCRC patients with the *BRAF* V600E mutation.

In the past, triple therapy with *BRAF*, MEK, and *EGRF* inhibitors has mostly been used in clinical studies in European and American populations, whereas the safety and efficacy of triple therapy in Asian populations have not been well reported. In December 2021, a single-center study was conducted on an Asian population [61]. Nine eligible mCRC patients with *BRAF* mutations were enrolled and received triple therapy, with a median follow-up of 14.5 months (1–26 months). A majority of the patients (88.8%) received two or more systemic treatments, with a triple therapy regimen consisting mainly of darafenib, trametinib, and panizumab. The overall response rate was 11.1% and the disease control rate was 33.3%. Adverse events were usually grade 1–2 and included nausea, hypertension, gastrointestinal symptoms, and skin disorders. Therefore, triple therapy for *BRAF*-mutated mCRC in the Asian population was concluded to be safe, well tolerated, and with good clinical efficacy. In January 2022, a single-center prospective clinical study administered mFOLFOXIRI + bevacizumab therapy (experimental group) or mFOLFOXIRI therapy alone (control group) as conversion therapy in patients with *RAS/BRAF/PIK3CA* mutations in initially unresectable CRLM [62]. The rate of patients with no evidence of disease (NED) was the primary endpoint. The NED rates in the experimental (54 cases) and control (26 cases) groups were 40.7% and 30.8%, respectively (*p* = 0.022). The overall response rates (ORR) of the experimental and control groups were 77.4% and 60.0%, respectively (*p* = 0.112). The median progression-free survival (PFS) in the experimental group was 12.6 months, which was higher than the PFS of 9.1 months for the control group, and the median OS in the experimental group was longer than that of the control group (42.6 months vs. 35.3 months, respectively, *p* = 0.052). Compared to mFOLFOXIRI alone, mFOLFOXIRI combined with bevacizumab increased the incidence of clinical NED and tended to improve survival. In another study (Visnu-2), which was a multicenter, randomized, phase II study [63], investigators studied the effect of *BRAF* and *PIK3CA* mutation status on first-line treatment with bevacizumab or cetuximab in combination with 5-fluorouracil/calcium folate and irinotecan (FOLFIRI) in patients with *RAS* wild-type mCRC. The findings suggested that *BRAF/PIK3CA* status affects the outcomes of patients with RAS wild-type mCRC, but does not appear to contribute to the selection of first-line targeted therapy.

With increasing research on the molecular mechanism of *BRAF* mutations, the idea of selecting a drug therapy based on the functional typing of *BRAF* mutations is increasingly being accepted. At present, clinical studies have mostly focused on V600E mutations given their high incidence, whereas relatively little attention has been paid to non-V600E mutations. A multicenter, retrospective study classified non-V600E BRAF mutations into different functional types based on signaling mechanisms and kinase activity: activated and *RAS* independent (Class 2) versus kinase-impaired and RAS dependent (Class 3) [64]. Class 2 *BRAF*-mutated mCRCs (n = 12) rarely responded to *EGFR* antibody therapy, whereas most Class 3 *BRAF*-mutated mCRCs (n = 28) responded to *EGFR* antibody therapy. Thus, *EGFR* monoclonal antibody therapy should be considered for CRC patients with Class 3 *BRAF* mutations.

## 6. Conclusions and Future Perspectives

*BRAF* mutation plays an important role in the development of CRC. V600E is a common and unique molecular subtype and accounts for approximately 80–90% of *BRAF* gene mutations. According to recent studies, *BRAF* mutations, and in particular the V600E subtype, predict poor outcomes in CRLM patients after hepatectomy and are associated with poorer OS and DFS. In fact, several studies have suggested that the OS of patients with resectable CRLM with *BRAF* mutations is as poor as that of patients with unresectable liver metastases. In addition, patients with *BRAF* V600E-mutated CRLM after metastasectomy are prone to early and high rates of recurrence that occur mostly within one year after surgical resection. However, other studies have suggested that the OS of patients with *BRAF* mutations after surgical resection is superior to that of patients undergoing systemic treatment, particularly for patients with non-V600E mutations. Thus, it remains unclear whether *BRAF* mutation and the V600E subtype should constitute a biologic contraindication. The FOLFOXIRI regimen (oxaliplatin + irinotecan + folate + fluorouracil) combined with bevacizumab remains the first-line treatment for *BRAF*-mutated mCRC. The safety, tolerability, and efficacy of triple therapy with *BRAF*, MEK, and *EGFR* inhibitors have been verified. However, the toxicity of triple therapy is still high and requires close clinical attention. Of note, the third class of RAS-dependent mutations in V600E *BRAF* responded well to monoclonal antibodies against *EGFR*. Thus, the early identification of such patients is particularly important for clinical decisions.

## Figures and Tables

**Figure 1 cancers-14-04067-f001:**
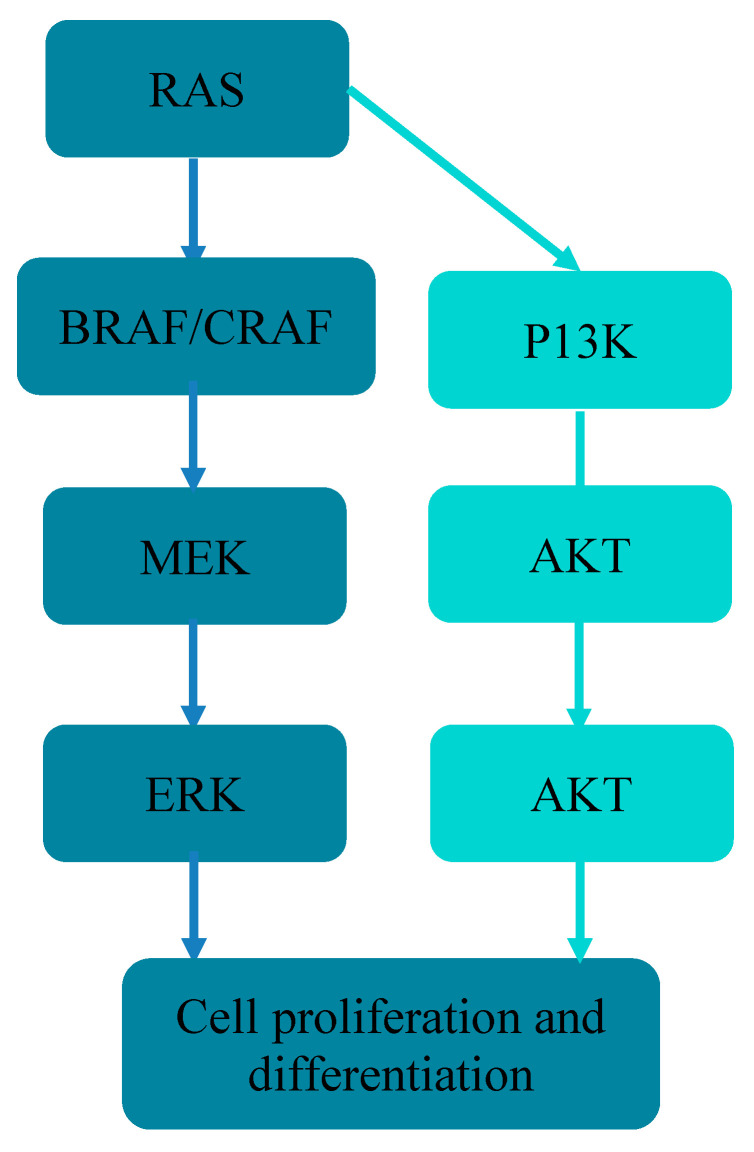
The molecular mechanism of BRAF and its activation of the MEK/ERK signaling and related down signaling pathways.

**Figure 2 cancers-14-04067-f002:**
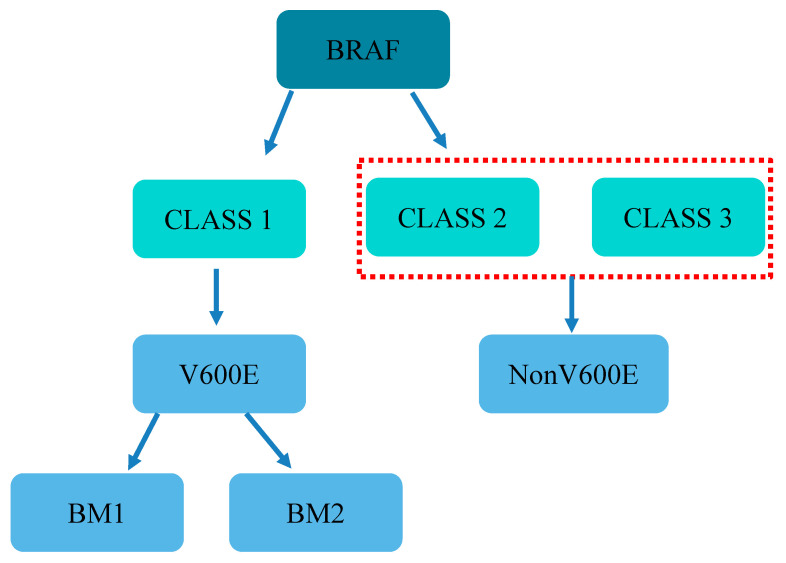
Classification and types of BRAF mutations.

**Table 1 cancers-14-04067-t001:** Summary of recent studies reporting survival outcomes of patients with BRAF mutation.

Study	N *	Research Type	BRAF Mutation (%)	Mutant Subtype	Overall Survival (OS)	Recurrence/Disease-Free Survival (RFS/DFS)
					HR (95%CI); *p*-Value	HR (95% CI) *p*-Value
Margonis et al., 2018 [13]	853	multicenter cohort study	41 (5.1%)	Yes	2.76 (1.74–4.37); *p* < 0.001	2.04 (1.30–3.20); *p* = 0.002
Bachet et al., 2019 [37]	249	case-matched study	66	No	NA; *p* = 0.004	1.16 (0.72–1.85); *p* = 0.547
Gagniere et al., 2020 [36]	1497	multicenter cohort study	35 (2%)	Yes	NA; *p* < 0.001	NA; *p* < 0.001
Yuan-Tzu et al., 2021 [39]	492	retrospective study	25 (5.1%)	No	NA; *p* = 0.006	NA
Shin et al., 2021 [42]	172	retrospective study	5 (2.9%)	No	27.6 (9.5–80.4); *p* < 0.001	12.5 (4.3–35.8); *p* < 0.001

**Table 2 cancers-14-04067-t002:** Toxicities of Targeted Therapies.

Targeted Therapeutics	Side Reactions			
BRAF inhibitor	arthralgia	Rash/allergic reaction	fatigue	hair loss
MEK inhibitor	diarrhea	photosensitized reaction	fever	hemorrhage
EGFR inhibitor	acne-like rash	diarrhea	allergic reaction	constipation
VEGR inhibitor	gastrointestinal perforation	wound healing complications	hemorrhage	hypertension

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
