# Peer review of "BRAF Mutations in Colorectal Liver Metastases: Prognostic Implications and Potential Therapeutic Strategies"

_cancers, 2022, doi:10.3390/cancers14174067_

Round 1

Reviewer 1 Report

I enjoyed this review article and I think it is very informative about current status of treatment for BRAF mutant colorectal cancer. My comments are as follows.

1)     4.3. Effect of surgical treatment on prognosis: Median OS of Japanese study (reference #38) was 31.1 months, and this study included only resected cases. French study (reference #42) was similar median OS of 34 months for resected cases. Unresected case was worse median OS of 10.6 months because initial resectability was different as 86% vs 24%. Therefore, there are big bias between these two groups and we cannot compare those. I could not find the reference of “another contemporary study in Japan” which indicated median OS 17.2 months. (Line 250). It means results of Japanese studies were also different. Please explain the possibility in order.

2)     (Line 305) Papillary thyroid cancer is high incidence of BRAF mutation, however it is one of the good prognosis cancers. I’m interested in the reason why it causes. Please comment about it.

Author Response

August 12, 2022 Re: “BRAF Mutations in Colorectal Liver Metastases: Prognostic Implications and Potential Therapeutic Strategies” Thank you for reviewing our manuscript. We are pleased that the manuscript was favorably reviewed and found to be potentially acceptable for publication pending revisions. We believe that the reviewers’ comments and the corresponding responses will serve to strengthen the manuscript. As requested, we have provided a point-by-point response to the comments below and a version of the manuscript with tracked changes clearly showing all additions and deletions.

Reviewer 2 Report

The authors have presented a well written review article on the therapeutic strategies for colorectal cancer patients with BRAF mutations. However there are some minor comments and queries I would like the authors to address before this could be published.

       1. In the section for targeted therapies the author should include carcinoembryonic antigen as one of important marker which is over expressed in varieties of cancers including the colorectal cancers and responsible for reduced overall survival after surgical resection of colorectal carcinoma (https://doi.org/10.1081/CNV-58878). These receptors have been proved to be useful for directing various targeted therapeutics in-vivo and in-vitro models (https://doi.org/10.1021/acsami.1c21655).

      2. Could the author discuss if BRAF mutation has any role to play with chemo-drug resistance in the CRC?

3    3. The authors should also highlight in their discussion the clinical significance of integrin family proteins as a biomarker in CRC patients. Integrins have been found to play as an important predictive marker of chemotherapy response in various cancer models as observed from clinical samples (https://doi.org/10.1038/s41416-021-01484-w)

      4. As the toxicity of the triple therapy using BRAF, MEK, and EGFR inhibitors against the BRAF-mutated mCRC is high, could the authors discuss how it could be improvised? Could a targeted therapeutic system to deliver the inhibitors resolve this issue?

      5. A simple flow chart of the molecular mechanism of BRAF and how it activates the MEK/ERK signalling and related down signalling pathways would be appreciated for inclusion in section 2.

Author Response

(The authors gave the same response as above.)
